# Exploring the Roles of Entrepreneurial Education, Proactive Personality and Creative Self-Efficacy in the Formation of Undergraduates’ New Venture Ideas in China

**DOI:** 10.3390/bs15020185

**Published:** 2025-02-10

**Authors:** Rui Hu, Caiyun Li

**Affiliations:** 1College of Education, Central China Normal University, Wuhan 430079, China; hurui@mail.hzau.edu.cn; 2School of Public Administration, Huazhong Agricultural University, Wuhan 430070, China; 3Department of Management, Wuhan College, Wuhan 430212, China

**Keywords:** entrepreneurial education, proactive personality, creative self-efficacy, new venture idea

## Abstract

A new venture idea is the prerequisite for discovering or building an entrepreneurial opportunity and also the foundation for undergraduates to start a new venture. Based on social cognitive theory, this paper explores the roles of entrepreneurial education, proactive personality, and creative self-efficacy in the formation of undergraduates’ new venture ideas. Using a sample of 650 undergraduates in China, we find that entrepreneurial education and proactive personality are both positively related to new venture idea formation, and proactive personalities also moderate the relationship between entrepreneurial education and new venture ideas. In addition, creative self-efficacy plays a mediating role in those direct effects mentioned above. That is, entrepreneurial education and a proactive personality can both promote the formation of undergraduates’ new venture ideas through creative self-efficacy. These results have practical significance for entrepreneurial education in universities, especially for fostering undergraduates’ creative self-efficacy.

## 1. Introduction

The core of entrepreneurship lies in entrepreneurial opportunity ([68]; [77]; [82]; [88]), but the idea, namely the new venture idea (NVI), is the prerequisite for the formation and development of entrepreneurial opportunity ([57]). For the individual seeking to identify or create a new entrepreneurial opportunity, initially there is an idea and a great deal of uncertainty ([68]; [30]). Before taking some actions to further develop this idea and reduce related uncertainties, it is still just an idea ([30]). If an idea is not developed, it will never become a formal opportunity. However, without an idea, opportunities will never emerge ([68]). The term “new venture idea” refers to the content, not how favorable the ideas are to what has previously been called opportunities, opportunity recognition, opportunity identification, or entrepreneurial discovery ([29]). It is an “imagined future venture” that represents what actors may want to create, rather than the gradual realization of the venture itself ([29]). [93] ([93]) defined a venture idea as a preliminary and mostly incomplete mental representation of the concept for a potential future venture, often containing vague insights into the potential for value creation.

Since 2014, the Chinese government has attached great importance to innovation and entrepreneurship, turning “mass entrepreneurship and innovation” into a new engine of economic growth in China. Subsequently, multiple policies related to entrepreneurial education were issued by the Chinese government and the Ministry of Education aiming to cultivate talents with innovative spirit, entrepreneurial awareness, and the ability to innovate and start businesses. Driven by those policies, entrepreneurial education is widely promoted in Chinese universities, including establishing entrepreneurship colleges, introducing entrepreneurship courses, promoting entrepreneurship competitions, creating entrepreneurship platforms, and providing other entrepreneurial support. Entrepreneurial education is regarded as an important breakthrough in higher education reform, dedicated to cultivating innovative and entrepreneurial talents needed for China’s economic transformation and growth. That is to say, whether to start a business or to get entrepreneurial success are not the criteria for measuring the effectiveness of entrepreneurial education; training undergraduates to have the courage to take risks, the willingness to innovate, and the ability to take action in entrepreneurship is more important ([69]). This is consistent with the connotation of the new venture idea, which is that generating more entrepreneurial ideas is more important than evaluating the quality of entrepreneurial ideas ([29]). Based on the long tail effect, only with a sufficient number of ideas emerging can high-quality breakthrough innovations potentially occur ([36]). Therefore, entrepreneurial education should aim to promote the formation of new venture ideas among undergraduates.

Previous research on new venture idea formation primarily focused on the business domain, encompassing new venture idea incubation and new venture opportunity evaluation ([70]), as well as the transition between venture ideas and venture formation ([48]). Recent scholarship is attentive to new venture ideation, which refers to the process of generating and developing new venture ideas ([73]). For the undergraduates, the main focus is on evaluating the effectiveness of entrepreneurial education. Numerous studies have confirmed the positive effect of entrepreneurial education on factors such as entrepreneurial motivation ([26]), entrepreneurial ability ([31]), entrepreneurial intention ([87]; [1]), and entrepreneurial behavior ([3]; [96]). In fact, the impact of entrepreneurial education on entrepreneurial activities has a lag effect. Especially true for undergraduates; the duration of the academic system is relatively short, making it difficult to effectively evaluate the commercial value of entrepreneurial opportunities. Moreover, it takes an even longer time to stimulate their entrepreneurial intentions and behaviors, which often occurs after graduation. Therefore, evaluating their NVIs may be more realistic. However, there is limited literature that elaborates on the formation mechanism of undergraduates’ NVIs ([94]).

Social cognitive theory holds that human activities are determined by the interaction of three factors: individual behavior, individual cognition and other individual characteristics, and the external environment in which the individual is located ([9]). Proactiveness, as one of the core characteristics of personality agency ([12]), can directly influence individual behavior and also interact with the environment to jointly influence individual behavior. Universities are important environmental factors that influencing undergraduates to engage in entrepreneurial activities, where the innovation atmosphere, entrepreneurial courses, and various entrepreneurial support are all components of entrepreneurial education ([38]). Therefore, proactive personality may not only directly affect the formation of undergraduates’ NVIs, but also interact with entrepreneurial education, the environment where undergraduates are located, to affect the formation of their NVIs. In addition, the core concept of social cognitive theory is self-efficacy, which refers to an individual’s subjective perception of their belief and ability to complete a specific activity ([10]). Individuals with high-level self-efficacy can generate novel and useful ideas through intrinsic drive ([37]). [11] ([11]) believes that self-efficacy originates from mastery experiences, vicarious experiences, social persuasion, and somatic and emotional states, so undergraduates’ self-efficacy can be enhanced through entrepreneurial education. Therefore, as a self-efficacy in the field of creative activities ([90]), creative self-efficacy may play an important role in the influence of entrepreneurial education on the formation of new venture ideas.

Based on the above discussion, this study intends to take social cognitive theory as the foundation, then introduce entrepreneurial education, proactive personality, and creative self-efficacy to address the shortcomings in the existing research on the formation mechanism of new venture ideas. The main objectives of this study are as follows: First, to verify the direct effects of entrepreneurial education and proactive personality on the formation of undergraduates’ new venture ideas. Second, to explore the moderating role of proactive personality in the impact of entrepreneurial education on new venture ideas. Third, to investigate the mediating role of creative self-efficacy in the relationship between entrepreneurial education and new venture ideas, and proactive personality and new venture ideas. The main contributions of this study are: First, to focus on the formation of undergraduates’ new venture ideas as the outcome variable of entrepreneurial education, rather than entrepreneurial intention or behavior as in previous researches, thus aligning more closely with the goals advocated by entrepreneurial education policies in China. Second, to delve deeply into the connotations of social cognitive theory, closely integrating the theoretical framework of this study, to make the research on the formation mechanism of new venture ideas more comprehensive and profound.

## 2. Literature Review

### 2.1. Entrepreneurial Education and New Venture Idea

In China, “entrepreneurial education” mainly refers to all kinds of entrepreneurial education courses, entrepreneurial activities, and entrepreneurial supports provided by universities. Entrepreneurial education is a type of learning activity aimed at enhancing entrepreneurial knowledge, abilities, skills, and individual characteristics ([28]). Entrepreneurial knowledge and skills gained through entrepreneurial education ([14]), contribute to create new connections and form new combinations that are technically feasible ([32]). They also help individuals understand data and forms that are novel, newly generated, and pattern-inconsistent, thereby leading them to follow with science, technology, and principles that facilitate the formation of creative ideas ([80]). In other words, entrepreneurial education improves undergraduates’ innovative consciousness and ability, encouraging them to break away from traditional thinking patterns and be brave enough to try and generate new ideas. Research indicates that new ideas on entrepreneurship are created outside the business context through experimentation in school teaching practices ([46]).

In addition, entrepreneurial education is conducive to improving the initiative and effectiveness of undergraduates in searching for entrepreneurial information, which is the key to the formation of new venture ideas. By creating an innovative and entrepreneurial campus atmosphere that encourages innovation and tolerates failure, entrepreneurial education greatly stimulates undergraduates’ entrepreneurial enthusiasm. Research has found that entrepreneurial education can help increase undergraduates’ willingness to engage in risk creation ([78]), and enhance their entrepreneurial mindset, such as innovation, motivation, and risk-taking ([72]). The stronger the willingness of undergraduates to take risks, the more likely they are to unleash their creative imagination and conduct entrepreneurial information searches, thus forming their own NVIs. Moreover, entrepreneurial education can also enhance their ability to identify entrepreneurial opportunity ([86]; [99]), that is, the comprehensive ability of entrepreneurs to search for information and perceive risks during the “opportunity window period” ([7]). The stronger the ability of undergraduates to search for entrepreneurial information, the more comprehensive the information they can obtain, which is more conducive to producing new venture ideas. Therefore, we proposed the following hypothesis:

**H1:** 
*Entrepreneurial education positively predicts undergraduates’ NVI.*


### 2.2. The Role of Proactive Personality

#### 2.2.1. Proactive Personality and the New Venture Idea

[13] ([13]) first proposed the concept of proactive personality, which refers to a relatively stable individual trait or behavioral tendency that is not hindered by external situational factors and actively takes action to transform or influence the external environment. [20] ([20]) summarized the common characteristics possessed by individuals with high-level proactive personality, including good professional knowledge and skills, proactive and courageous expression of their ideas, strong organizational commitment and sense of responsibility, and higher value pursuits. These characteristics are beneficial for the formation of new venture ideas, as reflected in the following: Firstly, individuals with a proactive personality are more likely to strive to improve their professional abilities, actively acquire relevant knowledge and skills needed for their own development, and thus exhibit stronger innovation tendency ([54]). Individuals with a high-level proactive personality tend to be self-starters, future-focused, and change-oriented ([74]). This positive expectation and belief in the future can stimulate individuals’ entrepreneurial enthusiasm and motivation, making them more likely to develop new venture ideas. Secondly, the proactive characteristics possessed by proactive personalities indicate an individual’s preference for risk and challenge, which is conducive to creativity ([54]; [56]). Some scholars even believe that proactive personality should be considered as a proxy indicator of personality traits that underlie creativity ([52]; [89]). Undergraduates with a proactive personality are more concerned about market changes and trends, and they are more likely to discover new business opportunities through observation and analysis, thus forming new venture ideas. In addition, individuals with proactive personality tend to set clear goals for achieving their expectations, and actively take action to achieve these goals. This goal-setting and action-orientation make undergraduates more likely to seek and conceive entrepreneurial projects actively ([27]). Thus, we proposed the following hypothesis:

**H2:** 
*Proactive personality positively predicts undergraduates’ NVI.*


#### 2.2.2. The Moderating Role of Proactive Personality

Social cognitive theory believes that individual differences are the products of a cognitive–affective system ([23]). That is to say, human beings are both products influenced by their environment and shapers of that environment ([10]). Individuals with a proactive personality are better at learning and communicating ([63]; [16]), which is beneficial for improving the effect of entrepreneurial education. Compared to the Big Five personality traits, proactive personality, as a compound personality trait, can better predict individuals’ motivation in learning contexts, thereby promoting their developmental activities ([66]). Compared to individuals with a low-level proactive personality, those with high-level tend to actively participate in knowledge-sharing activities and stimulate strong intrinsic motivation ([24]). Therefore, in the context of entrepreneurial education, individuals with a high-level proactive personality may be more actively engaged in entrepreneurial courses and practices, acquiring more entrepreneurial knowledge and skills, which contributes to the development of new venture ideas.

Moreover, individuals with a high-level proactive personality are more self-motivated in scanning their environment for opportunities ([75]). They are also more likely to actively change their environment, seek opportunities, or achieve better behavioral outcomes ([47]) and create and manage exchange relationships with their environment ([62]). They therefore benefit more from the entrepreneurial support provided by universities ([71]) and invest more effort in gathering information to generate new venture ideas ([19]). Therefore, proactive personality may interact with entrepreneurial education, which is the educational environment in which undergraduates are located, to affect the formation of new venture ideas together. Based on this, we proposed the following hypothesis:

**H3:** 
*Proactive personality moderates the relationship between entrepreneurial education and undergraduates’ NVI.*


### 2.3. The Role of Creative Self-Efficacy

Self-efficacy, as one of the core concepts of social cognitive theory, is a key factor in behavior change, achievement, and personal well-being ([10]). The importance of self-efficacy is particularly evident in tasks that require creativity and the discovery of new knowledge ([11]). [5] ([5]) defines creativity as the generation of novel and practical ideas or solutions, which is the initial step in the broader innovation process, and has been highlighted as an important element leading individuals to start their own ventures ([34]). Based on the above reasons, [90] ([90]) combined Bandura’s self-efficacy theory with Amabile’s creativity theory and then proposed creative self-efficacy, which refers to the confidence and assurance in one’s capacity to generate creative outcomes.

#### 2.3.1. The Mediating Role of Creative Self-Efficacy Between Entrepreneurial Education and New Venture Idea Formation

The positive impact of creative self-efficacy on employees’ creativity and creative activities ([91]; [42]), creative ideation ([64]; [79]), and undergraduates’ creative performance ([81]; [18]) have been all widely confirmed. Individuals high in creative self-efficacy possess confidence in their capacities to come up with innovative or creative ideas and implement them ([51]). These individuals are more willing to spend more time in the cognitive creative process to identify problems and propose new ideas to provide solutions for these problems ([83]). Creativity relates to an individual’s propensity to generate novel ideas and outcomes varying from pre-figured mental sets ([60]), which is consistent with the direction of creative generation of new venture ideas. Therefore, creative self-efficacy may also have a similar positive impact on new venture idea formation.

Creative self-efficacy is malleable and can be shaped through various forms of training and intervention ([92]). [15]’s ([15]) study showed that improving classroom atmosphere and teacher support behavior can enhance students’ creative self-efficacy. [67]’s ([67]) intervention study found that training can indeed improve individual creative self-efficacy and that a five-day training course is more effective than a one-day training course. Therefore, providing individuals with appropriate training to promote mastery experiences seems to increase their belief in their own abilities, which is consistent with the assumptions of social cognitive theory ([11]). Entrepreneurial education can create self-efficacy in individuals, the necessary skills and capabilities of which are indispensable for the initiation of a new business venture ([85]). Scholars also believe that participating in entrepreneurial courses that involve students in the creative process is an “environmental” input that shapes these individual cognitive processes and abilities ([17]; [98]). Based on this, we proposed the following hypothesis:

**H4:** 
*Creative self-efficacy mediates the relationship between entrepreneurial education and undergraduates’ NVI.*


#### 2.3.2. The Mediating Role of Creative Self-Efficacy Between Proactive Personality and the New Venture Idea

In addition, creative self-efficacy is also individual-specific, because personality traits directly affect creative self-efficacy ([35]). Initiative, as one of the core characteristics of a proactive personality ([12]), is closely related to self-efficacy and achievement needs ([55]). Because individuals with a proactive personality are self-starting and driven, they often exhibit high-level work involvement ([22]; [50]). In this process, individuals can accumulate mastery experiences and a sense of fulfillment, thereby enhancing their self-efficacy. Furthermore, proactive individuals show higher levels of creative self-efficacy, which fosters anticipatory cognition of entrepreneurship ([40]). They maintain an open attitude towards learning and growth ([2]), which makes them more positive in facing opportunities and challenges. This attitude of continuous learning and growth helps individuals accumulate more experiences and knowledge, thus enhancing their self-efficacy in different areas. Thus, we proposed the following hypothesis:

**H5:** 
*Creative self-efficacy mediates the relationship between proactive personality and undergraduates’ NVI.*


Based on the above discussion, this study aims to explore the mechanism of entrepreneurial education on undergraduates’ NVIs. The theoretical model is presented in Figure 1.

## 3. Method

### 3.1. Sample and Data Collection

To collect data on the relevant variables, we organized a questionnaire by translating well-established scales that were initially developed in English. A total of 1950 undergraduates who received entrepreneurial education participated anonymously and voluntarily in the survey with the help of school counselors from 21 universities in China through cluster random sampling. This sampling technique has been employed in Chinese higher education studies such as [76] ([76]). Ten of the universities are located in eastern China, eight universities are in Central China, and the other three are in western China. The school counselors distributed the web link to the electronic questionnaire on the Questionnaire Star platform to their class QQ or WeChat groups, informed students of the survey’s purpose and requirements, and encouraged them to answer it. Considering research ethics, the first question in the questionnaire is “Do you agree to fill in?”. If the participant chooses “Disagree”, it will end directly. Over two months, we collected data from 760 undergraduates, with 38.97% response rate. As Z. [65] ([65]) has pointed out, online surveys generally result in a lower response rate in China. A series of statistical assessments indicated that there was no significant difference between the responding and non-responding groups in terms of school region, student gender, age, etc. After deleting straight-lining and missing data, 650 valid questionnaires were retained, with an effective rate of 85.52%. The general information of the participants is as follows: 31% identify as male and 69% as female; 56% are rural residents; 58% are from Double First-Class Universities; 319 are majoring in natural sciences, 109 in business, and the other 222 in literature, art, education, and other humanities.

### 3.2. Measurement Instrument

New venture idea (NVI). We adopted [33]’s ([33]) scale to measure NVI (see in Appendix A). The authors claim that “opportunity discovery processes” refer to the initial conception and further development of a venture idea, which actually equates with NVI. The nine-item scale was mainly to check whether the participants has engaged in a deliberate search for venture ideas or actively communicated these ideas with others (e.g., “I have discussed ideas for a new business with my friends and family”), so the measures are formative. All responses to each item were scored 0 (No) or 1 (Yes); these responses to the nine items were then summed to form the index. This scale has been tested in the United States, mainland China, and Taiwan, which has shown good applicability. For evaluating its validity, a “redundancy analysis” was carried out. Following the recommendations of [95] ([95]) and [44] ([44]), we used a global item with 5-point Likert scale (1 = never; 2 = rarely; 3 = sometimes; 4 = often; and 5 = always) to reflect NVI frequency. This item was: “How often do you have new venture ideas during the past six months?”

Entrepreneurial education (EE). We used [38]’s ([38]) six-item scale to measure undergraduates’ perception of the entrepreneurial environment in their universities (e.g., “The courses foster the social and leadership skills needed by entrepreneurs”) (see in Appendix A). The rating scale for all items ranged from 1 (“strongly disagree”) to 5 (“strongly agree”). Here, the scale had a Cronbach’s α value of 0.909.

Proactive personality (PP). We measured proactive personality using [13]’s ([13]) ten measurement questions (e.g., “I am constantly on the lookout for new ways to improve my life”) (see in Appendix A). They were rated on a five-point Likert scale from 1 (“strongly disagree”) to 5 (“strongly agree”). This scale has been translated or adapted, and widely used in Chinese contexts ([49]). Here, the scale had a Cronbach’s α value of 0.908.

Creative self-efficacy (CSE). We used the three-item creative self-efficacy scale developed by [90] ([90]) (e.g., “I have confidence in my ability to solve problems creatively”) (see in Appendix A). Participants rated their agreement with each item on a five-point Likert scale from 1 (“strongly disagree”) to 5 (“strongly agree”). This scale has been widely used in higher education domains, such as in a Spain sample ([8]) and a Chinese sample ([43]). Here, the scale had a Cronbach’s α value of 0.850.

### 3.3. Data Analysis

Data analysis began with analyzing the demographic characteristics of the respondents and descriptive statistics using SPSS 25.0. An exploratory factor analysis was then performed to check common method bias. In order to test the hypotheses of this study, we employed a partial least squares structural equation modeling (PLS-SEM) approach to establish the relationships among entrepreneurial education, proactive personality, creative self-efficacy, and NVI. PLS-SEM was suitable for our study for two main reasons. Firstly, PLS-SEM is capable of handling formative and reflective variables, both of which were included in our model. Secondly, PLS-SEM is appropriate for prediction and theoretical extension ([44]). The proposed model was tested in two stages. The reliability and validity of the measurement model were initially examined following the guidelines by [44] ([44]). Afterwards, the structural model was estimated using the bootstrapping algorithm involving 5000 sub-samples. SmartPLS 4 was used for PLS-SEM analysis.

## 4. Results

### 4.1. Preliminary Analysis

First, the Harman single-factor test was conducted to examine whether there is a serious common method bias. A factor analysis was performed on all items, resulting in the extraction of 4 common factors, which explained 59.549% of the variance. The first common factor explained 35.729% of the total variance, which is less than the critical value of 40%. Therefore, there was no serious common method bias in this study.

Second, we examined the convergent validity, indicator weights and loadings, and indicator multicollinearity for NVI ([44]). For the convergent validity, we built a new model which used the formative latent variable for NVI to predict the reflective latent variable for that operationalized through the global item. The path coefficient between two latent variables was 0.875. As can be seen from Table 1, all indicator weights were statistically significant except NVI_5, NVI_7, and NVI_9. However, T-Values for these three loadings are bigger than 1.96 ([95]). To preserve the construct’s content validity, they were retained. Moreover, the VIF values in Table 1 are all smaller than 3.

Third, the indicator loadings, reliability, convergent validity, and discriminant validity of the three reflective measurement models were used in this study. As shown in Table 2, each indicator has a higher loading than 0.708 except for PP_3 and PP_6 of proactive personality. However, these two indicator loadings meet the criteria of 0.6 recommended by [25] ([25]). Also, [44] ([44]) stated that if value AVE is more than 0.5, the indicator with low loading values can be kept to maintain the content validity. Meanwhile, the composite reliabilities (CR) of different variables significantly exceed 0.70, and the average variance extracted (AVE) for each variable exceeds 0.50.

Table 3 shows a significant positive correlation between entrepreneurial education, proactive personality, creative self-efficacy, and NVI. In addition, the square root of the AVE for each variable in Table 3 is greater than the correlation coefficient between the variables. The Heterotrait-Monotrait Ratio is below 0.90 accepted as admissible, indicating good discriminant validity.

Lastly, a structural equation model was established. The significance test included p-values at the 0.05 level and 95% bootstrap confidence intervals. The model fit indices are presented in Table 4: the R^2^ for endogenous variables all exceed 0.3; VIF for exogenous variables and mediator are all less than 5; blindfolding-based Q^2^ is larger than 0; SRMR = 0.048; NFI = 0.831 (>0.8).

### 4.2. Hypotheses Test

First, the interaction effect of entrepreneurial education and proactive personality on NVI was examined. The results, as shown in Table 5, indicate that entrepreneurial education (β = 0.148, *p* < 0.001) and proactive personality (β = 0.064, *p* = 0.003) both have significant positive effects on NVI, and proactive personality moderates the relationship between entrepreneurial education and NVI (β = 0.054, *p* < 0.001). Therefore, H1, H2, and H3 are all validated.

Then, the mediating role of creative self-efficacy in the above-mentioned influence mechanism was examined. The results, as shown in Table 5, indicate that creative self-efficacy mediates the relationship between entrepreneurial education and NVI (β = 0.031, *p* = 0.001). It also mediates the relationship between proactive personality and NVI (β = 0.045, *p* = 0.001), so H4 and H5 are both validated.

## 5. Discussion

Firstly, the research findings indicate that creative self-efficacy plays a mediating role between entrepreneurial education and NVI. Entrepreneurial education can not only directly promote the formation of undergraduates’ NVIs, but also indirectly influence NVI formation through creative self-efficacy. These conclusions are consistent with the previous research that entrepreneurship has innate teachability ([45]). Entrepreneurial education can effectively deepen students’ understanding of entrepreneurial knowledge and skills, and guide their subsequent entrepreneurial activities ([14]). The NVI, as the foundation for undergraduates to engage in entrepreneurial activities and also the prerequisite for discovering or forming entrepreneurial opportunity, should be valued as one of the achievements of entrepreneurial education. In addition, entrepreneurial education can indirectly influence the formation of NVIs through creative self-efficacy, indicating that entrepreneurial education can develop undergraduates’ creative self-efficacy. By providing entrepreneurial practice activities that can accumulate alternative experiences for undergraduates, and inviting successful entrepreneurs to share their experiences, which can significantly improve undergraduates’ creative self-efficacy. This conclusion is also consistent with [11]’s ([11]) statement on the sources of self-efficacy. Previous research has found that entrepreneurial education can create self-efficacy in individuals and indirectly influence their entrepreneurial intentions ([78]; [6]; [97]). NVIs and entrepreneurial intention are both outcome variables of undergraduates’ entrepreneurship, influenced by self-efficacy.

Secondly, research has found that creative self-efficacy plays a mediating role between proactive personality and NVI formation. Proactive personality not only directly predicts NVI, but also indirectly influences NVI through creative self-efficacy. Individuals with a proactive personality are change-oriented and are not satisfied with passively adapting to their environment, but seek to actively improve or create new environments ([13]). This transformative orientation can stimulate individuals’ creative thinking and generate creative ideas. Studies found that employees with a proactive personality may display innovative behavior ([84]; [53]), which is consistent with our findings in this study that proactive personality positively influences NVI. In addition, proactive personality is closely related to self-efficacy and achievement needs ([55]). Individuals with a proactive personality are also adept at reflective learning from successful experiences ([61]), which is also one of the sources of self-efficacy as stated by Bandura. More importantly, individuals with proactive personality have positive expectations and beliefs about the future, which contribute to higher levels of NVI ([39]).

In addition, research has found that proactive personality plays a moderating role between entrepreneurial education and undergraduates’ NVI. Individuals with higher levels of proactive personality are more active in participating in entrepreneurial courses and practices, actively acquiring relevant knowledge and skills needed for their own development, and thus exhibiting stronger innovation tendency ([54]). On the contrary, individuals with lower levels of proactive personality have insufficient initiative and participation in entrepreneurial learning, which limits their creative thinking. In addition, individuals with higher levels of proactive personality are more sensitive to the entrepreneurial support and atmosphere provided by universities, and are more likely to identify and seize opportunities ([41]; [4]). As mentioned at the beginning, an opportunity is an idea with commercial value. Therefore, proactive personality may enhance the positive effect of entrepreneurial education on NVI formation, and even improve the quality of NVIs.

### 5.1. Theoretical Contribution

This study expands the effectiveness research of entrepreneurial education. Previous studies took entrepreneurial opportunity, entrepreneurial ability, entrepreneurial intention, and entrepreneurial behavior as outcome variables to explore the effectiveness of entrepreneurial education. The goal advocated by China’s entrepreneurial education policy is to develop undergraduates’ entrepreneurial spirit, awareness, and innovation and entrepreneurship abilities, rather than just entrepreneurial behavior. Therefore, this study takes NVI formation as the outcome variable, which is more in line with policy objectives. Moreover, due to the limitation of the 4-year undergraduate program in China, it is difficult to confirm the commercial value inherent in entrepreneurial opportunity in the short term. Therefore, research on NVI formation, which serves as the foundation for entrepreneurial activities, is more realistic.

Furthermore, this study enriches the research on the formation mechanism of NVIs among undergraduates. Previous studies have mainly focused on defining the concept of the NVI, especially distinguishing between NVIs and entrepreneurial opportunities based on the stages of entrepreneurial activities ([68]; [30]; [29]; [93]; [58]), as well as evaluating NVIs ([39]; [21]). There are few studies exploring the formation mechanism of NVIs, and empirical research is even more lacking. Moreover, most studies on the relationship between proactive personality and creativity, or creative self-efficacy and creativity, have focused on only the work and organizational field. Although proactive personality and creative self-efficacy have been discussed in the field of education, little research in this field has investigated their relationship with NVI.

In addition, this study deepens the application of social cognitive theory. By applying social cognitive theory to the field of entrepreneurial education, this study further validates and expands its role in explaining individual behavior, particularly in the formation process of NVIs. This study explores the mediating role of creative self-efficacy between entrepreneurial education and NVI formation, which helps us to understand how individuals can enhance their creative self-efficacy through entrepreneurial education, thereby forming NVIs and providing targeted intervention points for entrepreneurial education. This study also explores how proactive personality moderates the relationship between entrepreneurial education and NVIs, which can reveal the different responses of individuals with different levels of proactive personality in entrepreneurial education and provide theoretical basis for personalized entrepreneurial education.

### 5.2. Practical Implications

Universities should improve the construction of entrepreneurial education system, putting effort into fostering undergraduates’ creative self-efficacy ([40]). Based on [11]’s ([11]) explanation of the sources of self-efficacy, universities can make efforts such as the following: First, provide students with more entrepreneurial practice, allowing them to personally experience the entire process of entrepreneurship, which will enhance their confidence in their own abilities through successful experiences. Second, offer more chances for undergraduates to communicate and cooperate with mentors, entrepreneurs, and investors, from which they may gain more entrepreneurial opportunities and resources. Professional guidance and support can help enhance the entrepreneurial confidence of undergraduates. Third, invite successful entrepreneurs to share their stories, inspiring students to believe that they can achieve similar success. From the perspective of social persuasion, entrepreneurial teachers should encourage students to be brave enough to try and innovate, helping to form a good atmosphere that is tolerant of failure in universities. More importantly, they should attach importance to teaching students “how to learn” through problem-based learning, thereby developing a strong belief in their ability to learn ([59]). Individuals with high levels of learning self-efficacy are more likely to believe in their ability to cope with new challenges and learn new knowledge, which are benefit for improving creative self-efficacy, thereby promoting the formation of NVIs.

Universities should also pay attention to the individual differences of undergraduates in entrepreneurial education. First, carry out diversified teaching and learning methods based on undergraduates’ interests and goals, training their ability for independent learning and development. Individuals with a high-level proactive personality have some common characteristics, such as be active to express their ideas and pursuing higher value ([20]). Previous studies have found a significant positive correlation between proactive personality and students’ willingness to have their own companies ([27]); individuals with a high-level proactive personality often evince stronger entrepreneurial desires. Universities should provide students with high-level proactive personalities with opportunities to seek novel and challenging learning, encouraging them to think independently and explore new fields. For example, teachers can adopt teaching methods such as teamwork and collective discussions. In addition, they should encourage students to convert venture ideas into practical projects. Previous research has shown that individuals with a proactive personality are more eager to communicate and collaborate with other entrepreneurs. Universities should enrich entrepreneurial practices and entrepreneurial resources for them, including funding support, entrepreneurial networks, entrepreneurial guidance, etc.

### 5.3. Research Limitations and Perspectives

Although valuable conclusions have been obtained through the theoretical framework design and empirical analysis in this study, there are still some limitations that are worth noting. Firstly, we collected data, which is cross-sectional data, through self-report. Cross-sectional studies cannot capture the dynamic processes between variables, and self-reporting is susceptible to biases from personal emotions, attitudes, and cognitive biases, which can affect the accuracy of research results. In the future, other methods of data collection are needed for supplementation, such as longitudinal studies and experimental designs. Especially for the measurement of NVI, it is necessary to develop scales that are more suitable for Chinese undergraduates, or to comprehensively measure the quantity and quality of NVIs through experiments. Secondly, we chose creative self-efficacy as a mediator and proactive personality as a moderator in this study, but creative self-efficacy as individual cognition and proactive personality as an individual trait are both individual factors. According to social cognitive theory, future research should focus on the roles of other environmental factors and individual behavioral variables, then expand the research on the influence mechanism of entrepreneurial education on entrepreneurial activities.

## 6. Conclusions

An NVI is a prerequisite for undergraduates to engage in entrepreneurial activities and its connotation is highly consistent with China’s entrepreneurial education policy. Currently, entrepreneurial education is widely implemented in Chinese universities, but its effectiveness is not clearly evaluated. This study aims to explore the formation mechanism of NVIs among undergraduates, which has been largely unaddressed in previous research. Based on social cognitive theory, we discussed the roles of entrepreneurial education, proactive personality, and creative self-efficacy in the formation of NVIs. The results have confirmed the interaction effects of entrepreneurial education and proactive personality on NVI formation, and the mediating role of creative self-efficacy between entrepreneurial education and NVIs, as well as between proactive personality and NVIs, which fills the gap in the research on the formation mechanism of NVIs. These findings not only provide new empirical support for social cognitive theory, but also offer important reference for policymakers in entrepreneurial education when formulating relevant policies. Those conclusions drawn from this study emphasized the importance of proactive personality and creative self-efficacy in entrepreneurial education, which has guiding significance for how to promote undergraduates’ NVI formation. In addition, this study also provides new directions for future research, such as further exploring the relationships between entrepreneurial education and other variables in the entrepreneurship-related field, as well as their applicability in different contexts. In summary, this study has significant implications in both academic and practical domains, offering new perspectives and ideas for research and practice in entrepreneurship-related fields.

## Figures and Tables

**Figure 1 behavsci-15-00185-f001:**
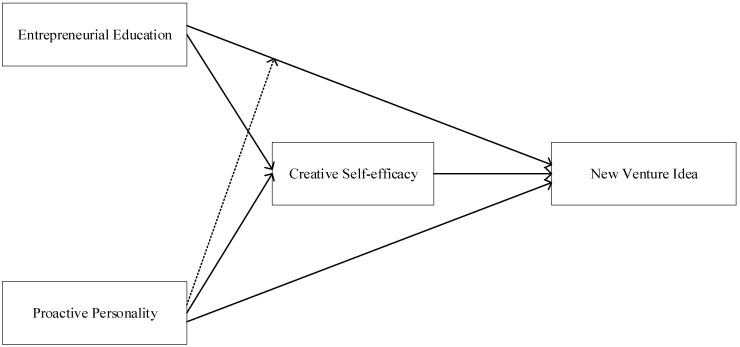
Theoretical Model.

**Table 1 behavsci-15-00185-t001:** Assessment of formative measurement model of new venture ideas.

Indicators	Outer Weights	*T*-Statistics*_weights_*	Outer Loadings	*T*-Statistics*_loadings_*	VIF
NVI_1	0.375	6.510	0.786	21.663	1.423
NVI_2	0.404	6.427	0.806	22.921	1.587
NVI_3	0.228	3.438	0.581	11.515	1.358
NVI_4	0.175	3.194	0.447	8.238	1.262
NVI_5	−0.036	0.691	0.365	6.518	1.299
NVI_6	0.197	3.785	0.348	5.705	1.031
NVI_7	0.046	0.750	0.376	7.153	1.235
NVI_8	0.207	3.789	0.427	7.693	1.207
NVI_9	0.016	0.234	0.359	7.318	1.241

**Table 2 behavsci-15-00185-t002:** Indicator loadings, reliability, and convergent validity of reflective measurement models.

Constructs	Indicators	Outer Loadings	*T*-Statistics	Cronbach Alpha (α)	CR(rho_a)	AVE
1. Entrepreneurial Education	EE_1	0.791	41.607	0.909	0.911	0.687
	EE_2	0.842	59.683			
	EE_3	0.846	61.105			
	EE_4	0.768	38.660			
	EE_5	0.855	65.705			
	EE_6	0.868	73.865			
2. Proactive Personality	PP_1	0.716	28.032	0.908	0.915	0.549
	PP_2	0.742	31.938			
	PP_3	0.640	19.137			
	PP_4	0.741	32.882			
	PP_5	0.803	47.930			
	PP_6	0.627	16.711			
	PP_7	0.742	33.234			
	PP_8	0.830	57.063			
	PP_9	0.779	45.906			
	PP_10	0.762	38.249			
3. Creative Self-efficacy	CSE_1	0.880	78.130	0.850	0.851	0.769
	CSE_2	0.902	85.262			
	CSE_3	0.848	53.783			

Note: N = 650. EE = Entrepreneurial Education, PP = Proactive Personality, CSE = Creative Self-efficacy, CR = composite reliability, AVE = average variance extracted.

**Table 3 behavsci-15-00185-t003:** Descriptive statistics and discriminant validity of the variables.

Constructs	M	SD	1	2	3	4	Heterotrait-Monotrait Ratio		
							1	2	3
1. Entrepreneurial Education	3.436	0.757	**0.829**						
2. Proactive Personality	3.208	0.547	0.836 ***	**0.741**			0.525		
3. Creative Self-efficacy	3.144	0.710	0.660 ***	0.792 ***	**0.877**		0.748	0.844	
4. New Venture Idea	2.020	1.840	0.529 ***	0.554 ***	0.519 ***	-			

Note: N = 650. *** *p* < 0.001. EE = Entrepreneurial Education, PP = Proactive Personality, M = mean, SD = standard deviation. The diagonal values in bold are square roots of AVE.

**Table 4 behavsci-15-00185-t004:** Results of structural model test.

Constructs	R^2^	Adj. R^2^	Q^2^	VIF	SRMR	NFI
Entrepreneurial Education	-	-	-	{1.299, 1.779}	-	-
Proactive Personality	-	-	-	{1.299, 2.314}	-	-
Creative Self-efficacy	0.684	0.683	0.522	3.172	-	-
New Venture Idea	0.398	0.394	0.101	-	0.048	0.831
Entrepreneurial Education × Proactive Personality	-	-	-	1.006	-	-

**Table 5 behavsci-15-00185-t005:** Results of hypotheses tests.

Structural Path	Path Coefficient/Indirect Effects	STDEV	95% CI		T Statistics	*p* Values	Conclusion
			LLCI	ULCI			
EE -> NVI	0.148	0.017	0.115	0.184	8.536	***	H1 is supported
PP -> NVI	0.064	0.022	0.021	0.107	2.937	0.003	H2 is supported
PP × EE -> NVI	0.054	0.007	0.041	0.070	7.442	***	H3 is supported
CSE -> NVI	0.080	0.024	0.032	0.127	3.334	0.001	H4 is supported
EE -> CSE	0.390	0.027	0.337	0.442	14.542	***	
EE -> CSE -> NVI	0.031	0.010	0.012	0.050	3.240	0.001	
PP -> CSE	0.566	0.025	0.516	0.616	22.526	***	H5 is supported
PP -> CSE -> NVI	0.045	0.014	0.018	0.073	3.275	0.001	

Note: *** *p* < 0.001.

## Data Availability

The datasets used and/or analysed during the current study are available from the corresponding author on reasonable request.

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
