# Peer review of "Exploring the Roles of Entrepreneurial Education, Proactive Personality and Creative Self-Efficacy in the Formation of Undergraduates’ New Venture Ideas in China"

_behavsci, 2025, doi:10.3390/bs15020185_

Round 1

Reviewer 1 Report

Comments and Suggestions for Authors

Very interesting research that contributes to business education.

Strengths of the article Relevance of the topic

The study addresses a current and significant issue: the impact of entrepreneurship education and psychological factors on the formation of new venture ideas among university students in China, aligning with national innovation and entrepreneurship policies.

Sound theoretical framework It is based on Bandura's social cognitive theory, which provides a robust theoretical framework for analyzing the interactions between individual and environmental factors.

Appropriate methodology Uses a considerable sample size (650 participants) and scales well established in previous studies, ensuring validity and reliability in the measurements. Significant results The findings confirm relevant relationships, such as the mediating role of creative self-efficacy and the moderating effect of proactive personality, which expands knowledge in the area.

Well-developed practical implications Provides clear recommendations for universities, such as designing personalized educational programs and strengthening support for students with proactive personalities.  

Weaknesses of the article Lack of updated references

Although key studies are included, several key references predate 2010. This could affect the perceived timeliness of the work.

Methodological limitations not addressed in depth. The research is based on cross-sectional data, which makes it difficult to establish relationships between the different studies.

Limited interpretation of marginal results Some indicators, such as NVI_5, NVI_7 and certain proactive personality items, show less significant coefficients, but their impact on the validity of the model is not sufficiently explained.

Conclusion  The conclusion lacks an emphatic closure that clearly summarizes the key contributions and potential impact of the study.

Author Response

Comments 1: Although key studies are included, several key references predate 2010. This could affect the perceived timeliness of the work.

Response 1: Thank you for pointing this out. We agree with this comment. Therefore, we have made modifications to the manuscript in two aspects. Firstly, in the third paragraph of the “Introduction” section, we added four recent NVI-related studies to highlight the limitations of existing research. Additionally, in the development of hypothesis, we included studies published after 2010, including two references in the development of H1 ([32][36]) and four references in the development of H3 ([48][50][52][54]).

Comments 2: Methodological limitations not addressed in depth. The research is based on cross-sectional data, which makes it difficult to establish relationships between the different studies.

Response 2: Agree. We acknowledge the methodological limitations of our study, particularly the use of cross-sectional data, which indeed makes it challenging to establish causal relationships between the variables of interest. It should be noted that the subjects of this study are undergraduates who have already received entrepreneurial education. This means that there is actually a certain time span between the entrepreneurial education and NVI’ formation.

We chose cross-sectional data due to the availability of comprehensive datasets that allowed us to examine a wide range of variables simultaneously. While this approach has provided valuable insights into the current state of the relationships we are studying, we recognize that it does not allow for the examination of temporal dynamics or causality. In response to this limitation, we have added a detailed discussion in the "Reasearch Limitations and Perspectives" section of our manuscript in lines 509-513, highlighting the potential impact of using cross-sectional data and suggesting that future research could benefit from longitudinal studies or experimental designs to better understand the causal pathways and temporal sequences involved.

Comments 3: Limited interpretation of marginal results. Some indicators, such as NVI_5, NVI_7 and certain proactive personality items, show less significant coefficients, but their impact on the validity of the model is not sufficiently explained.

Response 3: Thank you for pointing this out. We agree with this comment. Therefore, we have added a reference here to support the rationality of retaining these three indicators in line 297.

Wong, K. K. . (2013). Partial least squares structural equation modeling (pls-sem) techniques using smartpls. Marketing Bulletin, 24(1), 1-32.

Comments 4: The conclusion lacks an emphatic closure that clearly summarizes the key contributions and potential impact of the study.

Response 4: Thank you for pointing this out. We agree with this comment. Upon reflection, we realize that we did not sufficiently emphasize the significance of our findings and their broader implications. To address this issue, we have revised the “Conclusion” section of our manuscript as follows in lines 522-544:

1) We have restructured the conclusion to begin with a brief recap of the study's background and objectives, providing a context for our key contributions.

2) We have explicitly listed the main findings of our research, highlighting their importance in advancing theoretical support, policy formulation, and the optimization of entrepreneurial education.

3) We have added a strong concluding statement that underscores its contributions to the future research direction and the overall significance of our research.

Reviewer 2 Report

Comments and Suggestions for Authors

This manuscript explores an interesting research issue on new venture idea determinants in the context of Chinese Universities. However, to be publishable, it needs a major revision. The following comments will guide the effort of the author(s) to submit a revised version.

General comments:

1.       Please create a Literature Review section.

2.       The methodology section should be revised to clarify the sampling and data collection method

Specific comments:

1.       Introduction and Literature Review                                                                                                           - Please rewrite this section based on Shepherd and Wiklund (2020)

Shepherd, D. A., & Wiklund, J. (2020). Simple rules, templates, and heuristics! An attempt to deconstruct the craft of writing an entrepreneurship paper. Entrepreneurship Theory and Practice, 44(3), 371-390.

- Use a separate subsection to talk about the national context before hypotheses development.

2.       Methodology

- You are requested to present in detail the sampling and data collection method – what you have is not sufficient to validate your findings. In addition, you need to provide more information about your sample (34% of the sampled students are from other fields of study – please clarify).

- Please describe in more detail the data analysis (not analyses), see section 2.3

- Please provide notes in tables 2 and 3 about the indicators –

- Please provide your scales in Appendix.

- Please clarify lines 230-32, 239-41, 245-46, 251-52

Minor comments:

-          Please think about revising the title

-          Lines 56-59: provide more references

-          Lines 60-62: provide references

-          Line 158: do not use capital letter for social

-          Line 298: correct “hypotheses testing”

-          Line 441: correct “conclusion”

Author Response

Comments 1: Introduction and Literature Review       

- Please rewrite this section based on Shepherd and Wiklund (2020)

Shepherd, D. A., & Wiklund, J. (2020). Simple rules, templates, and heuristics! An attempt to deconstruct the craft of writing an entrepreneurship paper. Entrepreneurship Theory and Practice, 44(3), 371-390.

- Use a separate subsection to talk about the national context before hypotheses development.

Response 1: Thank you for pointing this out. We agree with this comment. After carefully reading the recommended literature, we have made the following revisions to our manuscript: 

1) We have divided the previous "Introduction" section into two parts: "Introduction" and "Literature Review." 

2) The "Introduction" section has also been reorganized based on the recommended literature. The first sentence of the first paragraph clearly indicates that this study belongs to entrepreneurship research, highlighting the importance of NVI in entrepreneurship studies. The second paragraph mainly introduces the national context, presenting the implementation and goals of entrepreneurial education in China from a practical perspective, and preliminarily showing the significant role of NVI in evaluating the effectiveness of entrepreneurial education. The third paragraph points out the gap in the research on the formation mechanism of NVI, especially the lack of studies on the impact of entrepreneurial education on NVI. The fourth paragraph remains unchanged, introducing Social cognitive theory to outline the theoretical framework of this study. 

3) In the last paragraph of the "Introduction," we have added the main objectives of this study and its contributions.

Comments 2: Methodology

- You are requested to present in detail the sampling and data collection method – what you have is not sufficient to validate your findings. In addition, you need to provide more information about your sample (34% of the sampled students are from other fields of study – please clarify).

-Please describe in more detail the data analysis (not analyses), see section 2.3

-Please provide notes in tables 2 and 3 about the indicators

-Please provide your scales in Appendix.

-Please clarify lines 230-32, 239-41, 245-46, 251-52

Response 2: Agree. Thank you very much for your valuable suggestions. We have further supplemented the sample composition and data collection process, including sample size, sampling method, questionnaire distribution method, and nonresponse bias results in lines 266-281. Additionally, we have added a reference in lines 277-278 to illustrate the real-world evidence of low response rates in online questionnaires. 

Liu, Z. ,  Hu, R. , &  Bi, X. . (2022). The effects of social media addiction on reading practice: a survey of undergraduate students in china. Journal of Documentation, 79, 670-682.

Besides, we have added notes in tables 2 and 3 about the indicators in lines 356-357 and 364-365, provide scales in Appendix in lines 555-591,and clarify all the details mentioned above.

Comments 3: Minor comments:

-Please think about revising the title

-Lines 56-59: provide more references

-Lines 60-62: provide references

-Line 158: do not use capital letter for social

-Line 298: correct “hypotheses testing”

-Line 441: correct “conclusion”

Response 3: Thank you for pointing this out. We agree with this comment. All the issues mentioned above have been modified.

Reviewer 3 Report

Comments and Suggestions for Authors

Title of the Article

The current title, while informative, could benefit from being more concise and precise. The phrase “How is …. Formed?” lacks the specificity needed in academic titles, particularly in terms of whether the study is exploring causal relationships or mediating effects.

 Suggestions:

The mediating role of creative self-efficacy on entrepreneurial education, proactive personality, and new venture idea formation in China.

Or

Exploring the role of new venture ideas in Chinese undergraduates: A study of entrepreneurial education proactive personality, and creative self-efficacy.

Theoretical background

The authors identify gaps in the literature, particularly concerning the formation of new venture ideas among undergraduate students in China. The article also clearly describes its objectives, contextualising them within the framework of proactive personality, social cognitive theory, and previous studies on entrepreneurial education.

However, the introduction of the section should be further enhanced by incorporating critiques of social cognitive theory and proactive personality in the entrepreneurial domain.

Lack of contextual factors

The article focuses on entrepreneurial education within Chinese universities without adequately addressing how cultural, institutional, or economic factors may influence their findings. A broader cultural or institutional lens would enrich the study’s applicability to diverse global contexts.

Research method
The research method and design are sound and well-communicated. The authors' use of established measurement scales and SEM further demonstrates rigour.

Discussion of Results and Findings

 The results are not only clearly presented, but also coherent and logically structured, providing the audience with a sense of being informed and enlightened. The findings were systematically linked to the hypotheses and theoretical model.

However, the article could benefit from more integrated narrative explanations of how these findings may translate into real-world applications. This would help the reader understand the research's practical implications.

Conclusion and Limitations

The conclusions are sound and well-supported by the data. The article integrates the findings with some practical implications.

The limitations discussed, while acknowledged, could be more thoroughly addressed in terms of practical solutions. For example, the authors acknowledge the constraints of cross-sectional design but could propose actionable recommendations for future studies to mitigate these limitations.

Author Response

Comments 1: The current title, while informative, could benefit from being more concise and precise. The phrase “How is …. Formed?” lacks the specificity needed in academic titles, particularly in terms of whether the study is exploring causal relationships or mediating effects.

 Suggestions:

The mediating role of creative self-efficacy on entrepreneurial education, proactive personality, and new venture idea formation in China.

Or: Exploring the role of new venture ideas in Chinese undergraduates: A study of entrepreneurial education proactive personality, and creative self-efficacy.

Response 1: Thank you for pointing this out. We agree with this comment. After careful consideration, we have decided to modify the title to: Exploring the roles of entrepreneurial education, proactive personality and creative self-efficacy in the formation of undergraduates’ new venture idea in China.

Comments 2: The introduction of the section should be further enhanced by incorporating critiques of social cognitive theory and proactive personality in the entrepreneurial domain.

Response 2: Thank you very much for your valuable suggestions. We partially agree with this comment and respond as follows:

First, in the third paragraph of the "Introduction," we have strengthened the discussion on the insufficiency of research on the formation mechanism of new venture idea, thereby highlighting the contributory role of Social cognitive theory in research on formation mechanism of new venture idea.

Second, regarding the Social cognitive theory in “Introduction,” there are some matters that need to be explained. Since the main purpose of introducing Social cognitive theory in the "Introduction" is to address the aforementioned research gap and then to introduce the moderating variable of proactive personality and the mediating variable of creative self-efficacy, we have conducted a concise theoretical analysis. The specific application of the theory will be more concretely reflected in the derivation of hypotheses where proactive personality serves as a moderating variable and creative self-efficacy serves as a mediating variable. Thank you again for your suggestions.

Comments 3: Lack of contextual factors.

The article focuses on entrepreneurial education within Chinese universities without adequately addressing how cultural, institutional, or economic factors may influence their findings. A broader cultural or institutional lens would enrich the study’s applicability to diverse global contexts.

Response 3: Thank you for pointing this out. We agree with this comment. Therefore, we have made meticulous revisions to the “Introduction” section. In the second paragraph of the “Introduction,” we have provided a detailed description to the background and objectives of entrepreneurial education in China, as well as the existing problems in the evaluation of entrepreneurial education. This leads to the rationality of using new venture ideas as an indicator of the effectiveness of entrepreneurial education, thereby highlighting the practical significance of this study.

Comments 4: The article could benefit from more integrated narrative explanations of how these findings may translate into real-world applications. This would help the reader understand the research's practical implications.

Response 4: Thank you for pointing this out. We agree with this comment. To make the specific measures clearer for readers, we have changed the logical structure of the presentation. We start with numbered headings to introduce the main sentences, which are the corresponding measures for universities, and then support them with existing literature.

Specifically, regarding "fostering undergraduates' creative self-efficacy," the measures are mainly based on "Bandura's (1997) explanation of the sources of self-efficacy" and are proposed from four aspects.

Regarding "implementing personalized entrepreneurship education," the measures mainly involve enriching teaching and learning methods based on individual characteristics, as well as providing entrepreneurial support for students with proactive personality.

Comments 5: The authors acknowledge the constraints of cross-sectional design but could propose actionable recommendations for future studies to mitigate these limitations.

Response 5: Thank you for pointing this out. We agree with this comment. In “Research Limitations and Perspectives”, we have added a discussion on the issues that may arise from the cross-sectional design, and have proposed ways to improve in the future, such as longitudinal studies and experimental designs in lines 509-513.

Reviewer 4 Report

Comments and Suggestions for Authors

Generally, a good paper but the authors should address some few issues:

Firstly, the authors should have separate headings leading to the development of each hypothesis. 

Secondly, the authors should provide more evidence to support the arguments leading to the development of H3.

Thirdly, on the Methods section, the statement 'Based on the principle of voluntariness' should be supported with a justification in literature. For example, to what extent is the sampling technique consistent with previous studies?

Under the Results section, the authors should provide evidence to support this statement, 'However, these two indicator loadings meet the criteria of 0.6'. This is because the 0.708 seems to be the generally accepted creteria, though some authors are okay with the 0.6. Hence, providing the appropriate justification is very important. 

Comments on the Quality of English Language

A few grammatical issues; 'Scholars generally agrees' at the Introduction section

Author Response

Comments 1: The authors should have separate headings leading to the development of each hypothesis.

Response 1: Thank you for pointing this out. We agree with this comment. In order to make the structure of the paper clearer and easier for readers to understand the process and basis for each hypothesis, we have added subheadings before the derivation of each hypothesis, such as "2.2.1. Proactive Personality and NVI" and "2.2.2. The Moderating Role of Proactive Personality."

Comments 2: The authors should provide more evidence to support the arguments leading to the development of H3.

Response 2: Agree. Thank you very much for the valuable suggestions. First, we have added six references to support the development of H3 in lines 181-197 ([48][50][52][53][54][55]). These references focus on how high-level proactive personality enhance entrepreneurial learning effectiveness and how high-level proactive personality affect the perception and utilization of entrepreneurial support. Furthermore, to make the reasoning logic clearer, we have divided the above two layers of explanation into two separate paragraphs.

Comments 3: On the Methods section, the statement 'Based on the principle of voluntariness' should be supported with a justification in literature. For example, to what extent is the sampling technique consistent with previous studies?

Response 3: Thank you for pointing this out. First, regarding the "the principle of voluntariness," the expression might not be accurate. What we intended to convey is that we ensured the voluntariness of participants' involvement in the survey through the design of the questions. This approach is consistent with the method used by Zhuang et al. (2020). Therefore, we have adjusted the wording in lines 272-276.

Zhuang, T., Cheung, A. C., & Tam, W. (2020). Modeling undergraduate STEM students’ satisfaction with their programs in China: an empirical study. Asia Pacific Education Review, 21, 211-225.

Second, concerning "the sampling technique," we have added a reference to Peng et al. (2022) to illustrate the rationality of using cluster random sampling in this study in line 270.

Peng, P., Ao, Y., Li, M., Wang, Y., Wang, T., & Bahmani, H. (2022). Building Information Modeling Learning Behavior of AEC Undergraduate Students in China. Behavioral Sciences, 12(8), 269.

Comments 4: Under the Results section, the authors should provide evidence to support this statement, 'However, these two indicator loadings meet the criteria of 0.6'. This is because the 0.708 seems to be the generally accepted creteria, though some authors are okay with the 0.6. Hence, providing the appropriate justification is very important. 

Response 4: Thank you for pointing this out. We agree with this comment. We have provided two references to illustrate the rationality of retaining these two indicators in lines 350-352.

Chin, W. W. (1998). The partial least squares approach to structural equation modeling. Modern methods for business research/Lawrence Erlbaum Associates. Modern Methods for Business Research, 295(2): 295-336.

Hair, J. F. ,  Hult, G. T. M. ,  Ringle, C. M. , &  Sarstedt, M. . (2017). A Primer on Partial Least Squares Structural Equation Modeling (PLS-SEM), 2nd edition. Sage Publications.
